# Synergistic Inhibition of Methicillin-Resistant *Staphylococcus aureus* (MRSA) by *Melaleuca alternifolia* Chell (Tea Tree) and *Eucalyptus globulus* Labill. Essential Oils in Association with Oxacillin

**DOI:** 10.3390/antibiotics12050846

**Published:** 2023-05-03

**Authors:** Ramona Iseppi, Carla Condò, Patrizia Messi

**Affiliations:** Department of Life Sciences, University of Modena and Reggio Emilia, Via Giuseppe Campi 287, 41125 Modena, Italy

**Keywords:** *Staphylococcus aureus*, essential oils, antibiotic, methicillin-resistant *Staphylococcus aureus*, *Melaleuca alternifolia*, *Eucalyptus globulus*

## Abstract

The presence of antibiotic-resistant bacteria has become a major therapeutic priority. This trend indicates the need for alternative agents to antibiotics, such as natural compounds of plant origin. By assessing membrane permeability, we investigated the antimicrobial activity of *Melaleuca alternifolia* and *Eucalyptus globulus* essential oils (EOs) against three strains of methicillin-resistant *Staphylococcus aureus* (MRSA). Using the checkerboard method, the efficacy of single EOs, in association with each other or in combination with oxacillin, was quantified by calculating the fractional inhibitory concentrations (FIC Index). All EOs showed a reduction in bacterial load, an alteration of membrane permeability which leads to an increase in its function, resulting in the release of nucleic acids and proteins. The treatment with EO–oxacillin combinations and associated EO–EO resulted in a synergistic effect in most of the tests performed. EO–EO association showed a high activity in the alteration of the membrane, increasing the permeability to about 80% in all the MRSA strains treated. In conclusion, the combination of EOs and antibiotics represents a valid therapeutic support against MRSA bacteria, allowing for a decrease in the antibiotic concentration needed for therapeutic use.

## 1. Introduction

Recently, bacteria resistant to multiple pharmacological drugs have steadily increased. The emergence of antibiotic-resistant bacteria (ARB) has become a major global problem [1], with a significant clinical and economic impact. The main causes of antibiotic resistance lie in the abuse of antibiotics for clinical therapy, and in the strong use in the veterinary and agricultural fields. Hence, the development of bacterial resistance to all classes of antibiotics leads to the continuous need to produce new drugs to combat it. In this scenario, the hypothesis of using non-chemotherapy treatments, such as photodynamic and photothermal therapy, has been explored [2,3]. All-in-one CO-gas-therapy-based versatile hydrogel dressing [4], and the development of nanotechnological methods to improve the penetrability of therapeutic agents is a promising manner for treating ARB bacterial infections [5]. Additionally, the hypothesis of using natural substances from plant sources, capable of limiting the diffusion and the transmission of these highly virulent microorganisms, has been proposed and confirmed by numerous investigations. Essential oils (EOs) are volatile, fragrant liquids, extracted from different parts of plants, with a long history of employment in folk medicine for the prevention and treatment of bacterial infections [6]. The antibacterial activity of EOs is commonly attributed to the perturbation of the structural integrity of the cell membrane, leading to bacterial cell death [7]. Their activity differs with the type of the target microorganism: Gram-positive bacteria are more sensitive than Gram-negative ones, due to their difference in cell wall composition and structure [8]. EOs can be used reliably, and its components can be employed as adjuvants in antimicrobial drugs to contrast the development of antibacterial, antifungal, and antiviral drugs resistance in pathogenic strains [9], according to the Food and Drug Administration (FDA) (2005) [10]. The exploitation of the synergistic combination of EOs and antimicrobial agents appears to be a possible alternative strategy to fight antimicrobial resistance. Some studies show the synergy between EOs and traditional antibiotics with consequent improvement of their efficacy [1,11,12,13,14] against ARB bacteria. Today, some plant-based antibacterial additives, such as essential oils and plant extracts, alone or in combination with antibiotics, have an application in wound dressings, such as skin lotions, plasters, hydrogels, and nanoemulsions [15].

*Melaleuca alternifolia* EO (Tea Tree Oil), an Australian native plant, is currently widely used as a topical antiseptic and anti-inflammatory agent in both cosmetic and personal care products [16,17]. *M. alternifolia* EO is known for its antibacterial properties [18,19], as it inhibits bacterial respiration, leading to the disruption of the permeability barrier of microbial membrane structures, and an induction of potassium ion leakage [20,21,22]. The antibacterial action of *M. alternifolia* is not carried out via a single mechanism, but also through a multicomponent action that mainly affects the cell wall [23]. Similarly, the phytochemical studies conducted on *Eucalyptus globulus* EO reveal that its biological activity is due to the high content of 1,8-cineole, a monoterpene, also present in *M. alternifolia* EO, together with terpinen-4-ol and to a-terpineol [24]. The *E. globulus* EO is more active against Gram-negative than Gram-positive bacteria, which is attributed to the presence of monoterpenes, such as 1,8-cineole, p-cymene and terpinolene, that can alter the permeability of the cytoplasmic membrane [25]. Its proper mechanism of antibacterial activity is still unknown, but many concomitant factors have been proposed, such as the capability to destroy cell wall and membranes, leading to the loss of ATP and metabolites [26,27].

Among ARB pathogens, methicillin-resistant *Staphylococcus aureus* (MRSA) is increased in many parts of the world, causing diseases often associated to a high mortality and morbidity rates [28,29]. The presence of many virulence factors enables MRSA strains to cause healthcare-associated disease, as bloodstream and catheter-related infections, and MRSA outbreaks can occur when a patient or healthcare worker without symptoms is colonized by an MRSA strain. The transmission of the pathogen can take place through both direct (spreading the strain to another person) and indirect contact (contaminated environment and surfaces). Similarly, MRSA strains determine community-associated infections, especially skin and soft tissue infections, and necrotizing pneumonia, in susceptible hosts. The mechanism of methicillin resistance in *S. aureus* is due to the expression of PC1 b-lactamase and the acquisition of the *mecA* gene encoding an altered penicillin-binding protein called PBP2′, with a low or no affinity for β-lactam antibiotics [30,31].

The aim of the present investigation was to find the best associations among *Melaleuca alternifolia* and *Eucalyptus globulus* EOs that result in significant antibacterial activity towards methicillin-resistant *Staphylococcus aureus* (MRSA) strains at low concentrations. Many in vitro and in vivo studies have already documented the biological benefits of EOs in this context; however, despite their promising antimicrobial properties and therapeutic potential, some issues are being raised about their use, the main one being the toxicity to mammalian cell membrane due to the lipophilic character of the EO compounds [32]. Therefore, the search for the best combinations capable of guaranteeing both antibacterial capability and low concentrations of use is an important objective of pursuit to overcome this problem. Similarly, the rapid emergence of antibiotic resistance worldwide has led to an urgent search for solutions that can overcome the concern of treatment failure related to ARB pathogens. Even in this case, many studies have shown the potential of EOs to mitigate antimicrobial resistance, as an alternative to existing antibiotics. The best EO–oxacillin combinations have been studied against MRSA strains for this purpose, while maintaining the goal of reducing the use concentrations of both the active compounds. Lastly, to confirm that the mode of action remained the same despite the mixtures used in the study, the ability to damage the structural integrity of the cell membrane of MRSA strains in planktonic states of the best associations/combinations has been studied. 

## 2. Results

### 2.1. Qualitative and Semi-Quantitative Analysis of Essential Oils

Using gas chromatography (GC), gas chromatography coupled with mass spectrometry (GC-MS), and gas chromatography coupled with mass spectrometry (GC-MS), *M. alternifolia* and *E. globulus* EOs were phytochemically characterized. 

The *M. alternifolia* EO displayed a composition rich in terpinen-4-ol (43.29%), γ-terpinene (20.16%), and α-terpinene (8.89%). Other phytochemical constituents were terpinolene (3.35%), α-terpineol (2.99%), p-cymene (2.84%), α-pinene (2.56%), and 1,8-cineole (2.35%). The *E. globulus* EO showed a composition in which the main components were 1,8-cineole (58.07%), linalool (12.05%), linalyl acetate (10.95%), camphor (4.39%), and α-pinene (2.33%) (Appendix A) [33]. 

### 2.2. Antibacterial Susceptibility and Fractional Inhibitory (FIC) Index Determination

The oxacillin resistance was confirmed for all MRSA strains, for both MIC values, ranging from 64 to 256 µg/mL, and the presence of the resistance gene *mecA* was determined for all the strains via polymerase chain reaction (PCR) (Appendix A). The *M. alternifolia* EO showed better antibacterial activity compared to the *E. globulus* EO, with MIC values ranging from 4 to 8 µg/mL and 32 to 256 µg/mL, respectively (Table 1).

With regard to the effectiveness of the natural compounds used in association (EO–EO) or in combination (EO–antibiotic), the most active synergies toward two MRSA strains (*S. aureus* 13 and 20) were observed for the oxacillin–*E. globulus* EO combination and associated EO–EO (FIC = 0.19), followed by the oxacillin–*M. alternifolia* EO combination. Regarding *S. aureus* 32, all the tested mixtures showed the same activity, with a fractional inhibitory concentration (FIC) of 0.5 (Table 2).

### 2.3. Growth Kinetics Study via Fluorescence Assay

In accordance with the MIC data of all tested strains, oxacillin did not show a concentration-dependent effect, with an absence of bacterial growth.

Concerning *S. aureus* 13 and *S. aureus* 20, the single EOs and all the tested mixtures showed a significant reduction in the bacterial load (*p* < 0.0001) from the initial contact times (Figure 1 and Figure 2). At 12 h of experimentation, *E. globulus* EO displayed a decrease in the bacterial load of *S. aureus* 32 (*p* = 0.0002), compared to the *M. alternifolia* EO and oxacillin–Eos combinations, and associated EO–EO.

At 16 h of experimentation and onward, single EOs and all the mixtures determined a marked reduction in *S. aureus* 32 cells count (*p* < 0.0001) (Figure 3).

### 2.4. Determination of Membrane Permeability Alteration via a Cristal Violet Assay

Crystal violet uptake by *S. aureus* 13 increased up to 80% when treated with associated EO–EO (*p* < 0.0001). High crystal violet absorption values of 74% and 67% have also been observed in presence of oxacillin–*M. alternifolia* EO and oxacillin–*E. globulus* EO combinations, respectively (*p* < 0.0001).

The *M. alternifolia* EO showed a greater activity against *S. aureus* 13 compared to the *E. globulus* EO, with crystal violet uptake values of 53.30% and 25.70%, respectively (*p* < 0.0001 and *p* = 0.0146).

Treatment of the strain with oxacillin exhibited an uptake value of 19.30%, a value very similar to the negative control (*S. aureus* 13 untreated), indicating no involvement of the antibiotic in the alteration of membrane permeability (Figure 4).

The uptake of crystal violet by *S. aureus* 20 was 53.60% when treated with the *M. alternifolia* EO (*p* = 0.0002), while it was 23.30% when exposed to the *E. globulus* EO.

The use of oxacillin–*M. alternifolia* EO and oxacillin–*E. globulus* EO combinations resulted in higher crystal violet uptake rates, which were 75% and 73.80%, respectively (*p* < 0.0001).

The associated EO–EO led to the best result, with an absorption value of 87% (*p* < 0.0001). Moreover, for *S. aureus* 20, treatment with oxacillin showed a low uptake value (20.60%), which indicates no alteration of the membrane permeability (Figure 5).

Regarding *S. aureus* 32, high crystal violet uptake values were observed after treatment with the oxacillin–*M. alternifolia* EO (67%) and oxacillin–*E. globulus* EO (73.17%) combinations (*p* < 0.0001), and with associated *M. alternifolia*–*E. globulus* EO (78%) (*p* < 0.0001).

When the EOs were used alone, good uptake values were observed, very similar to each other (54.6% and 53% for *M. alternifolia* EO and *E. globulus* EO, respectively).

Once again, treatment with oxacillin determined a low uptake value (21.60%), which indicates no alteration of the membrane permeability (Figure 6).

### 2.5. Membrane Permeability Test via an Addition of SDS 0.1%

Treatment with oxacillin showed an absence of membrane permeability through the influx of SDS into all treated MRSA cells (Figure 7, Figure 8 and Figure 9). In relation to *S. aureus* 13, membrane permeabilization as a result of treatment with *M. alternifolia* EO, oxacillin–*M. alternifolia* EO, and *M. alternifolia* EO–*E. globulus* EO mixtures was observed after 10 min (*p* = 0.0028, *p* = 0.0042 and *p* = 0.0010, respectively). The *E. globulus* EO alone did not show any activity after 30 min of treatment, while at the end of the experiment, it displayed a good alteration in permeability (*p* = 0.0001).

At the end of the test (60 min), all EOs, alone and in combinations, were able to alter membrane permeability, with the best activity shown by the *M. alternifolia* EO, alone and in all mixtures (*p* < 0.0001) (Figure 7).

Activity targeting the membrane of *S. aureus* 20 was achieved after 30 min of treatment, with good activity results for the associated EO–EO (*p* = 0.0005).

After 60 min, all compounds (EOs, EO–EO, and oxacillin–EO) determined the permeability of *S. aureus* 20 cell membranes (*p* < 0.0001) (Figure 8).

For *S. aureus* 32, the membrane permeability activity was evidenced after 30 min of exposure. The association *M. alternifolia* EO–*E. globulus* EO showed the best activity (*p* = 0.0009), which was maintained until the end of the experiment (*p* < 0.0001).

The antibiotic–EO combinations also displayed excellent membrane permeabilization (*p* < 0.0001), but only after 60 min of treatment (Figure 9).

### 2.6. Release of Nucleic Acids and Proteins

Figure 10, Figure 11 and Figure 12 display the optical density of filtrate from the three *S. aureus* MRSA strains, untreated or treated with single EOs, oxacillin, and all mixtures (EO–EO and antibiotic–EO) measured at 260 and 280 nm. The rise in optical density at 260 nm demonstrated an increase in nucleic acids, while the rise at 280 nm indicated an increase in proteins released from the bacterial cells. Treatment with oxacillin showed an absence of the release of nucleic acids and proteins into all treated MRSA cells (Figure 10, Figure 11 and Figure 12).

Figure 10a shows that the release of nucleic acids (260 nm) from *S. aureus* 13 was significantly higher for all treatments, particularly when the strain was treated with the *M. alternifolia* EO (*p* < 0.0001) and *M. alternifolia* EO–*E. globulus* EO (*p* < 0.0001). The protein release (280 nm) was higher when *S. aureus* 13 was treated with the oxacillin–*M. alternifolia* EO (*p* < 0.0001), oxacillin–*E. globulus* EO (*p* < 0.0001), and EO–EO (*p* < 0.0001) mixtures (Figure 10b).

The release of nucleic acids (260 nm) from *S. aureus* 20 was significantly higher for all treatments, except the oxacillin–*E. globulus* EO combination (Figure 11a). Regarding protein release (Figure 11b), treatments with the *E. globulus* EO alone and oxacillin–EO combinations showed a low activity. On the contrary, the *M. alternifolia* EO (*p* = 0.0053) and *M. alternifolia* EO–*E. globulus* EO (*p* = 0.0004) showed a high activity, which allowed for the release of proteins from *S. aureus* 20.

For *S. aureus* 32, a release of nucleic acids was observed following all treatments, particularly the oxacillin–EO and EO–EO mixtures (*p* < 0.0001) (Figure 12a).

However, no protein release (280 nm) was observed following treatment with the *E. globulus* EO, and there was a low release for the oxacillin–*M. alternifolia* EO combination (*p* = 0.0266). Following treatments with the *M. alternifolia* EO and oxacillin–*E. globulus* EO, a protein release was obtained (*p* = 0.0031 and *p* = 0.0043, respectively), and even more so with the *M. alternifolia* EO–*E. globulus* EO treatment (*p* = 0.0001) (Figure 12b).

### 2.7. Scanning Electron Microscopy

Scanning electron microscopy images showed the morphological changes in *S. aureus* strains when treated with single EOs and EO–EO combinations added at the MIC and fractional inhibitory concentration index levels. The treated cells displayed irregularities compared to the original spherical shape, while the untreated samples presented spherical, regular, and intact cells (Figure 13).

## 3. Discussion

Infectious diseases have long been considered a global public health priority due to their major health and economic impact on the population. Over the years, the overuse of antibiotics has led to the proliferation of infections by bacterial strains resistant to some classes of antibiotics, becoming a major concern worldwide, as the type of resistance to these antibiotics greatly limits the effective options for the treatment of infected patients. For this reason, studies have been carried out for some years on the diffusion of this phenomenon, along with research aimed at identifying strategies and new antibacterial compounds capable of counteracting this global health phenomenon.

Among the invasive isolates of the eight key bacterial species under surveillance by the EARS-Net, a network that collect data on antimicrobial resistance (AMR) provided by 29 European Union (EU) and European Economic Area (EEA) countries in 2021 (data referring to 2020), *S. aureus* represents the second (21.9%) most reported bacterial species in Europe (ECDC, 2022) [29]. In a recent study, it was estimated that *S. aureus* with resistance to methicillin caused more than 100,000 deaths [34]. Although during the 2016–2020 period a decreasing trend was observed, MRSA remains an important pathogen in the EU/EEA, with levels remaining high across several countries. In the present investigation, three MRSA strains out of 35 *S. aureus* strains have been isolated (8.70%), all harboring the *mecA* gene, which encodes the low-affinity penicillin-binding protein, PBP 2A [35], and is widely disseminated within the *S. aureus* population [36]. The isolation of oxacillin-susceptible *mecA*-positive *S. aureus* strains is increasingly being reported; however studies have recently found that the *mecC* gene also mediates drug resistance development [37]. Polymerase chain reaction (PCR) detection of the *mecA* or *mecC* genes is used as a gold standard test for MRSA status.

The problem of the AMR world-spread also fits into the broader context of the ONE HEALTH approach, that recognizes the interconnection between people, animals, plants, and their shared environment. It is widely known that the health of people is closely connected to the health of animals and our shared environment. From an epidemiological point of view, strains of MRSA can be divided into three broad categories: community- associated (CA), healthcare-associated (HA), and livestock-associated (LA) MRSA. Although HA MRSA and CA MRSA strains mainly affect humans, LA MRSA has been detected not only in farm animals, but also in people whose working activity involves close contact with animals (vets, breeders, etc.), with a carriage of 3.90% in EU, apart from five countries with higher isolation rates [38]. The changes in MRSA epidemiology are also shown by the emergence of livestock-associated MRSA (LA MRSA) strains within the human population [39]. Lastly, complete resistance to vancomycin has emerged within the past two decades, with the first vancomycin-resistant *S. aureus* strain isolated in 2002 in the USA, followed by reports of VRSA recovery in other countries, including the European area [40,41]. Vancomycin is one of the first-line drugs for the treatment of MRSA binfections, and the resistance of *S. aureus* to this antibiotic is a concern for the failure of therapy in infectious diseases caused by these resistant strains. Resistance to vancomycin in VRSA *S. aureus* is mediated by a *vanA* gene cluster, which is transferred from vancomycin-resistant enterococcus, as already demonstrated in studies carried out both in vitro and in mice [42,43].

Therefore, studies on the spread and, even more, on the type of MRSA strains within the population are essential, both to describe epidemiological trends and find effective infection control strategies, including the development of new drugs of natural origin able to contribute to the reduction in deaths associated or attributable to these infections. In the present investigation, we found that the *M. alternifolia* EO and *E. globulus* EO act directly on the cell membrane, destroying the structure, and thus increasing its permeability in the planktonic state. This leads to the consequent loss of the basic structural functions of bacteria and, consequently, to bacterial cell death. The *M. alternifolia* EO and *E. globulus* EO displayed a composition rich in monoterpenes and phenolic compounds, that can act in synergy with each other by influencing the integrity of the bacterial membrane [44,45,46,47].

Our results showed a decrease in viable cells of all three methicillin-resistant MRSA isolates when treated with single EOs and all the combinations (EO–EO and oxacillin–EO). In general, the association *M. alternifolia* EO–*E. globulus* EO displayed a relevant activity by destroying the integrity of cell membranes, increasing their permeability with an enhancement in leakage of nucleic acids, proteins, and changes in bacterial morphology. In addition to the demonstration of the mode of action of the EOs, this study not only highlights the association of several essential oils or the combination of the same with the reference antibiotic led to cell damage and, consequently, to the death of the bacterium, but also that the mixtures can greatly reduce the concentrations of therapeutic use of all the compounds. Some studies show that the combination of EOs with antibiotics can reduce bacterial resistance and broaden the antimicrobial spectrum. Furthermore, the EO–antibiotic combination reduces the concentration of the drugs, reducing their toxicity [48,49,50]. The employment of EOs in the prevention of bacterial resistance is a very promising strategy, because many traditional antibiotics are pure compounds with only one target site of action, while essential oils composed of multiple active compounds can act at different levels [51]. Today, essential oils and plant extracts, alone or in combination with antibiotics, have an application in wound dressings, chosen based on their healing and antibacterial properties [52]. Some studies indicate the use of essential oils in hydrocolloids, foams, films, dermal patches, and electrospun polymer dressings to enhance the wound healing process [53] and improve skin penetrability [54]. Several methods have been used to incorporate essential oils into dressings, such as the modification of commercial dressings by immersion in solutions of the active compound, or their encapsulation in the polymeric fibers which constitute the dressings [55].

Hence, the combination of EOs and antibiotics represents a valid therapeutic alternative against MRSA bacteria, allowing a decrease in the antibiotic concentration needed for therapeutic use.

## 4. Materials and Methods

### 4.1. Microbial Strains and Essential Oils

Three clinical isolates of MRSA *Staphylococcus aureus*, isolated from patients affected by bacteriemia, were confirmed via matrix-assisted laser desorption ionization (MALDI) time-of-flight mass spectrometry (TOF/MS). The strains were initially screened using BD BBL CHROMagar MRSA II (Becton Dickinson GmbH, Heidelberg, Germany), a selective chromogenic agar medium, to determine their resistance to oxacillin, and were stored at the microbiology laboratory (University of Modena and Reggio Emilia, Italy) at −80 °C, in media containing 20% (*w*/*v*) glycerol, until use.

The *Melaleuca alternifolia* Chell (Tea-tree) and *Eucalyptus globulus* Labill. EOs were purchased from a local herbalist shop in Modena, Italy. These EOs were chosen for their better antibacterial capacity that has been established in a previous investigation [22]. The chemical characterization via gas chromatography-mass spectrometry (GC/MS) and gas chromatography with flame ionization detection (GC-FID) techniques for the two EOs purchased and obtained via hydrodistillation, has been described in a previous study [33].

### 4.2. Antimicrobial Susceptibility Testing and Detection of Methicillin-Resistant Staphylococcus aureus (MRSA) Genes

The in vitro activity of oxacillin was determined using a broth microdilution method in 96-well microplates, according to the Clinical Laboratory Standards Institute (CLSI) guidelines, 2019 [56]. Briefly, 95 µL of nutrient broth and 5 µL of cell suspensions were dispensed into each well, up to final inoculum concentrations of 10^6^ CFU/mL. Then, 100 μL of serial dilutions of oxacillin were added to obtain concentrations ranging from 512 to 0.25 μg/mL. The last well, containing 195 μL of nutrient broth and 5 μL of antibiotic-free test strains, was used as a negative control. The plates were incubated at 37 °C for 24 h, mixed on a plate shaker at 300 rpm for 20 s, and the MIC was defined as the lowest concentration of oxacillin that inhibited the visible growth of the test organisms. The optical density (OD) was then measured at 570 nm, using a microplate reader. All experiments were repeated three times.

The presence of the resistance gene, *mecA*, was confirmed for all strains via PCR. DNA, used as a template during the PCR, was obtained from bacterial colonies for each strain tested via a simple boiling method [57]. The primer sequence is presented in Table 3.

### 4.3. Determination of MIC (Minimum Inhibitory Concentration) and FIC Index (Fractional Inhibitory Concentration)

To determine the MIC of Eos, the broth microdilution method was employed, following the Clinical and Laboratory Standards Institute (CLSI) guidelines, 2019 [56]. Briefly, sterile 96-well microplates, each containing 95 µL of tryptic soy broth (TSB) (Oxoid S.p.A, Milan, Italy), were supplemented with 5 µL of bacterial suspensions (10^6^ CFU/mL) and 100 µL of EO serial dilutions (from 512 to 0.125 μg/mL). After incubation at 37 °C for 24 h, MIC values were obtained by optical density (OD) at 570 nm.

To check the synergistic antibacterial activity of associated EO–EO and EO–antibiotic combinations, the fractional inhibitory concentration (FIC) index was studied against MRSA strains, in the same way as previously described for the MIC evaluation. The FIC index determination was carried out via a checkerboard method [48], and the values were calculated by comparing the MIC of each agent alone with the MIC of the respective association or combination. The fractional inhibitory concentration (FIC) index was calculated using the formula FIC = MIC (oxacillin–*M. alternifolia*)/MIC oxacillin + MIC (oxacillin–*M. alternifolia*)/MIC *M. alternifolia* EO, FIC = MIC (oxacillin–*E. globulus*)/MIC oxacillin + MIC (oxacillin–*E. globulus*)/MIC *E. globulus* EO, and FIC = MIC (*M. alternifolia* EO–*E. globulus* EO)/MIC *M. alternifolia* EO + MIC (*M. alternifolia* EO–*E. globulus* EO)/MIC *E. globulus* EO. An FIC index of ≤0.5 reveals synergism, ≤0.5 to ≥1 an additive effect, 1 to 4 indifference, and >4 antagonism.

### 4.4. Growth Kinetics Study via Fluorescence Assay

The growth of all the test strains was determined in the presence of EOs and oxacillin by themselves or in combination (EO–EO and antimicrobial–EO) added at the MIC and fractional inhibitory concentration index levels. The broth cultures were diluted overnight to obtain a density of about 10^2^ CFU/mL, and were placed in a 96-well sterile microplate, along with 90 μL of sterile nutrient broth and 10 μL of the strains from a stock, previously diluted. The microplate was incubated at 37 °C, and the treated and untreated wells (control) were stained by the “live/dead cells stain kit” (Thermo Fisher Scientific, Waltham, MA, USA) at predetermined time intervals (0, 12, 18, and 24 h), according to manufacturer instructions. The method is based on the use 5(6)-carboxyfluorescein diacetate (cFDA) to detect alive cells. The fluorescence emission of CFDA (excitation/emission: 485/528) added to each well (10 μL) was analyzed using a multi-well fluorescence plate reader (Synergy HTX, BIOTEK, Winooski, VT, USA). The results were expressed as relative fluorescence units (RFUs).

### 4.5. Determination of Membrane Permeability Alteration via a Cristal Violet Assay

Alteration of membrane permeability was detected via a crystal violet assay [58]. Bacterial suspensions were grown in 2 mL of TSB at 37 °C for 24 h, and then centrifuged at 4500× *g* for 5 min at 4 °C, washed twice and resuspended in PBS (Phosphate Buffer Saline, pH-7.4). The EOs and oxacillin were then added at the respective MIC of each strain, and for both association EO–EO and combination EO–oxacillin at the synergistic concentrations, as previously detected by the FIC index determination. The suspensions were incubated at 37 °C for 30 min. Bacterial suspensions without EOs and oxacillin were used as a negative control. The cells were centrifuged at 9300× *g* for 5 min and resuspended in PBS containing 10 µg/mL of crystal violet. The crystal-violet-treated suspensions were incubated for 10 min at 37 °C, centrifuged at 13,400× *g* for 15 min, and their optical density (OD) was measured at 590 nm using a microplate reader (Sunrise Tecan, Austria) as an arithmetic mean of the three determinations, and the standard deviation was reported as error bars. The percentage of crystal violet absorbed was calculated using the following formula: (OD value of sample)/(OD value of pure crystal violet solution) × 100.

### 4.6. Membrane Permeability Test via an Addition of SDS 0.1%

The quantification of bacterial membrane permeability by single compounds and their associations was determined through the combined use of the methods of Marri et al. [59], with some modifications. Overnight cultures were centrifuged at 4000× *g* for 15 min, resuspended in 5 mL of PBS, then centrifuged again at 4000× *g* for 15 min and resuspended in PBS. The pellet obtained for each strain was treated with the different concentrations of single compounds and their associations, based on the previously obtained MICs and FIC indexes. After incubation at 37 °C for 24 h, the samples were washed twice in PBS via centrifugation at 14,000× *g* for 10 min at 4 °C. The supernatants were removed, and 1 mL of PBS was added. The samples were divided into two aliquots: a part of each sample was inoculated into a 96-well sterile microplate, and 100 µL of sodium-dodecyl–sulfate (SDS) were added at the final concentration of 0.1%. Aliquots without an SDS were used as a negative control. The alteration of the bacterial membrane caused by a sudden efflux was determined by detecting a decrease in OD at 595 nm at different intervals (time 0, 10, 30, and 60 min) using a microplate reader (Sunrise Tecan, Austria).

### 4.7. Release of Nucleic Acids and Proteins

The release of nucleic acids and proteins of all the test strains was determined in the presence of oxacillin, single EOs or in combination (EO–EO and antimicrobial–EO) added at the MIC and fractional inhibitory concentration index levels, according to Moghimi et al. [60] with some modifications. Ten µL of an overnight culture was inoculated into a sterile tube containing the different concentrations of single compounds and their associations, and incubated at room temperature for 16–18 h. Then, a 1 mL aliquot of each sample was taken and filtered with a 0.45 µm filter. The absorbance of the filtrates was measured at 260 and 280 nm using a spectrophotometer (GloMax^®^ Discover, Promega Corporation, Madison, WI, USA). The absorbance of the control samples without culture addition was subtracted from the absorbance of the respectively treated samples. Experiments were performed in triplicate and the results are expressed as the mean optical densities of nucleic acids (260 nm) and proteins (280 nm).

### 4.8. Scanning Electron Microscopy

Scanning electron microscopy (SEM) was used to observe the morphological changes of *S. aureus* after exposure to the single EOs and at the association EO–EO for 16 h at 37 °C. Thereafter, the strains were washed with PBS and centrifuged at 4000× *g* for 15 min at 4 °C to collect the bacterial pellets. The pellet obtained for each strain was washed with PBS and fixed in 2.5% glutaraldehyde at 4 °C for 6 h. Then, the cells were dehydrated with serial increasing concentrations of ethanol (25%, 50%, 75%, 95%, and 100%). Finally, the samples were sputter-coated with gold, and observed via SEM (FEI Nova NanoSEM™, Hillsboro, OR, USA).

### 4.9. Statistical Analysis

All experiments were replicated three times. The statistical analysis was performed using a t-test and an ANOVA test with a statistical program GraphPad Prism 9.2.0 (San Diego, CA, USA). Values of *p* ≤ 0.05 were considered statistically significant.

## 5. Conclusions

The presence of antibiotic-resistant pathogens has become a major problem with a significant impact involving not only human health, but also animal health and their shared environment. Within this scenario, studies on the use of natural substances capable of positively modulating the sensitivity of pathogens resistant to antibiotics are included. The data from our study indicate that the *Malaleuca alternifolia* and *Eucalyptus globulus* EOs are able to permeabilize the membrane of methicillin-resistant *Staphylococcus aureus* (MRSA) strains.

The cell membranes of *S. aureus* showed irreversible damage confirmed by increased permeability of the cell membrane, increased leakage of nucleic acids, proteins and changes in bacterial morphology after treatment not only with the single EOs, but also when combined with oxacillin and in association with each other.

The activity of the oxacillin–EO combination against MRSA at lower concentrations could be an expression of a restored susceptibility to the reference antibiotic induced by EO, with important clinical implications. These results could provide further information for a possible role of the *M. alternifolia* EO and *E. globulus* EO as viable alternatives to address the problem of MRSA and also other pathogens, such as pandrug-resistant Gram-negative bacteria.

## Figures and Tables

**Figure 1 antibiotics-12-00846-f001:**
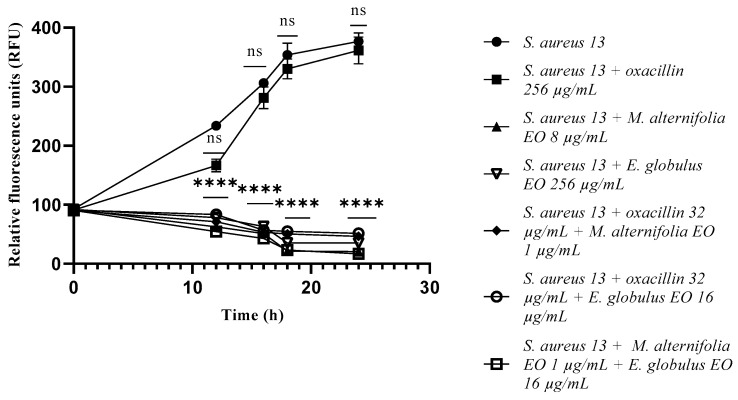
Growth of methicillin-resistant *Staphylococcus aureus* (MRSA) 13 cells detected via a fluorescence assay. Each experiment was replicated three times. *p*-values of <0.05 (*), *p* < 0.01 (**), *p* < 0.001 (***) and *p* < 0.0001 (****) were considered significant according to a *t*-test and an ANOVA. ns stands for not statistically significant.

**Figure 2 antibiotics-12-00846-f002:**
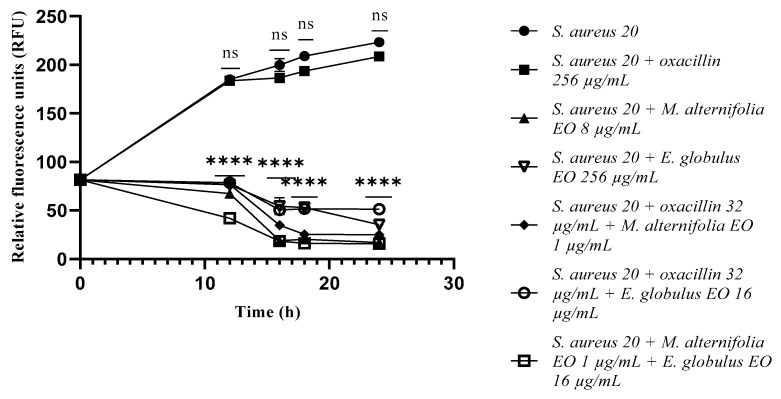
Growth of methicillin-resistant *Staphylococcus aureus* (MRSA) 20 cells detected via a fluorescence assay. Each experiment was replicated three times. *p*-values of <0.05 (*), *p* < 0.01 (**), *p* < 0.001 (***) and *p* < 0.0001 (****) were considered significant according to a *t*-test and an ANOVA. ns stands for not statistically significant.

**Figure 3 antibiotics-12-00846-f003:**
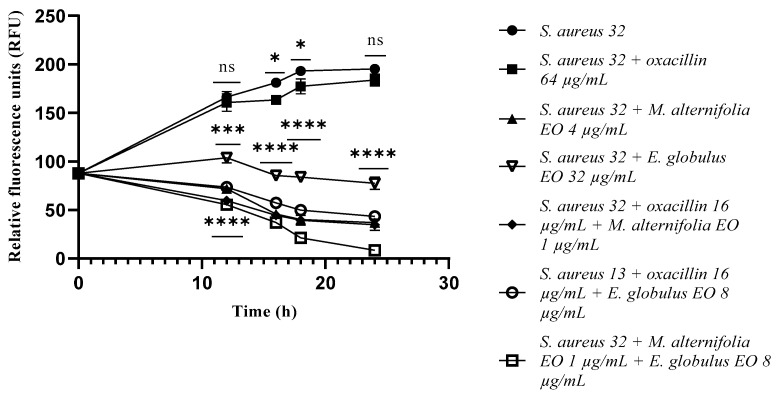
Growth of methicillin-resistant *Staphylococcus aureus* (MRSA) 32 cells detected via a fluorescence assay. Each experiment was replicated three times. *p*-values of <0.05 (*), *p* < 0.01 (**), *p* < 0.001 (***) and *p* < 0.0001 (****) were considered significant according to a *t*-test and an ANOVA. ns stands for not statistically significant.

**Figure 4 antibiotics-12-00846-f004:**
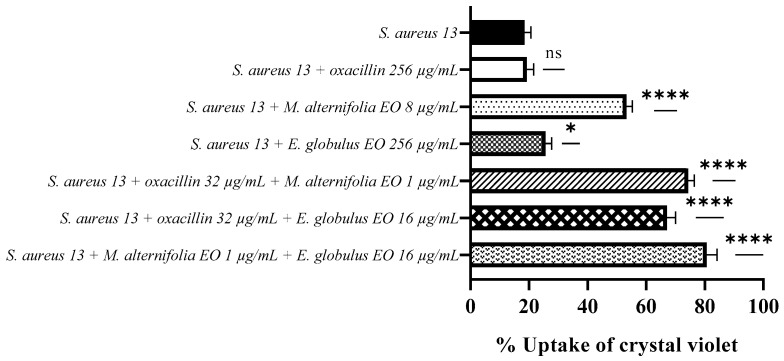
Measurement of cell permeability detected via a crystal violet assay for methicillin-resistant *Staphylococcus aureus* (MRSA) 13. Each experiment was replicated three times. *p*-values of <0.05 (*), *p* < 0.01 (**), *p* < 0.001 (***) and *p* < 0.0001 (****) were considered significant according to a *t*-test and an ANOVA. ns stands for not statistically significant.

**Figure 5 antibiotics-12-00846-f005:**
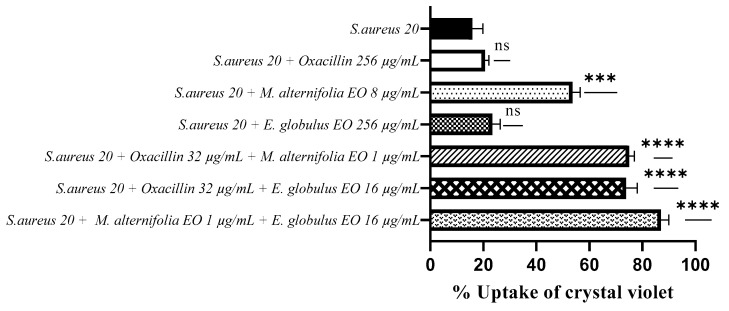
Measurement of cell permeability detected via a crystal violet assay for methicillin-resistant *Staphylococcus aureus* (MRSA) 20. Each experiment was replicated three times. *p*-values of <0.05 (*), *p* < 0.01 (**), *p* < 0.001 (***) and *p* < 0.0001 (****) were considered significant according to a *t*-test and an ANOVA. ns stands for not statistically significant.

**Figure 6 antibiotics-12-00846-f006:**
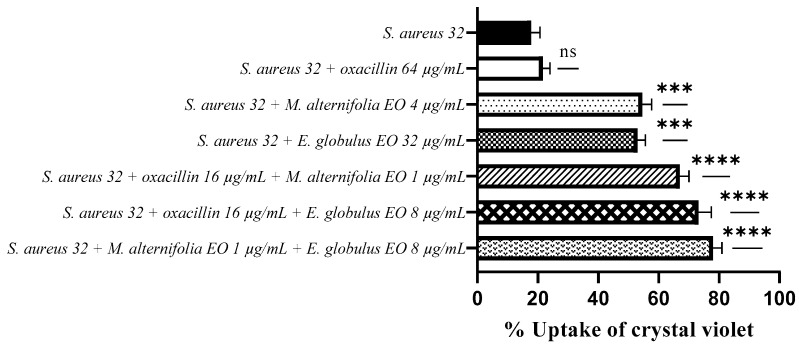
Measurement of cell permeability detected via a crystal violet assay for methicillin-resistant *Staphylococcus aureus* (MRSA) 32. Each experiment was replicated three times. *p*-values of <0.05 (*), *p* < 0.01 (**), *p* < 0.001 (***) and *p* < 0.0001 (****) were considered significant according to a *t*-test and an ANOVA. ns stands for not statistically significant.

**Figure 7 antibiotics-12-00846-f007:**
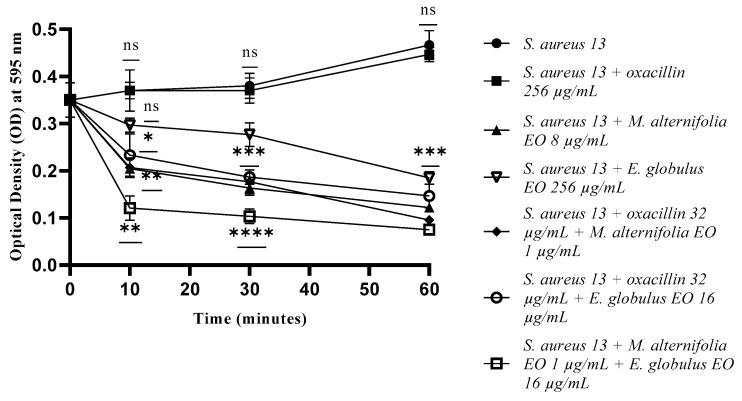
Membrane permeability test via an addition of SDS 0.1% of *Staphylococcus aureus* (MRSA) 13 cells detected by addition of SDS 0.1%. Each experiment was replicated three times. *p*-values of <0.05 (*), *p* < 0.01 (**), *p* < 0.001 (***) and *p* < 0.0001 (****) were considered significant according to a *t*-test and an ANOVA. ns stands for not statistically significant.

**Figure 8 antibiotics-12-00846-f008:**
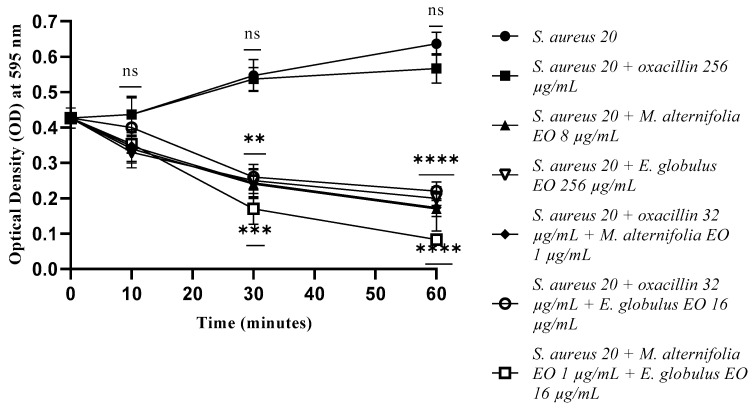
Membrane permeability test via an addition of SDS 0.1% of *Staphylococcus aureus* (MRSA) 20 cells detected by addition of SDS 0.1%. Each experiment was replicated three times. *p*-values of <0.05 (*), *p* < 0.01 (**), *p* < 0.001 (***) and *p* < 0.0001 (****) were considered significant according to a *t*-test and an ANOVA. ns stands for not statistically significant.

**Figure 9 antibiotics-12-00846-f009:**
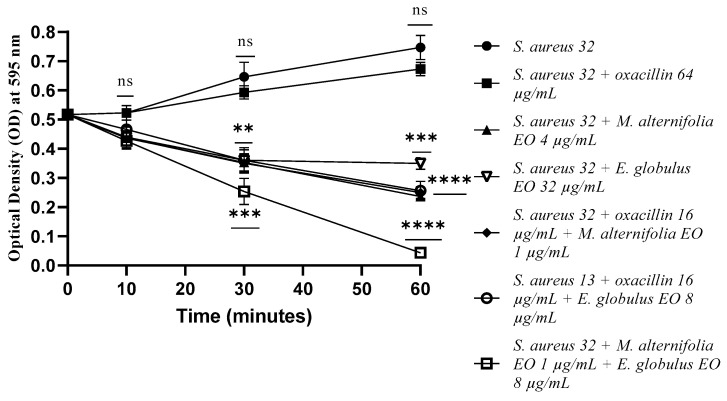
Membrane permeability test via an addition of SDS 0.1% of *Staphylococcus aureus* (MRSA) 32 cells detected by addition of SDS 0.1%. Each experiment was replicated three times. *p*-values of <0.05 (*), *p* < 0.01 (**), *p* < 0.001 (***) and *p* < 0.0001 (****) were considered significant according to a *t*-test and an ANOVA. ns stands for not statistically significant.

**Figure 10 antibiotics-12-00846-f010:**
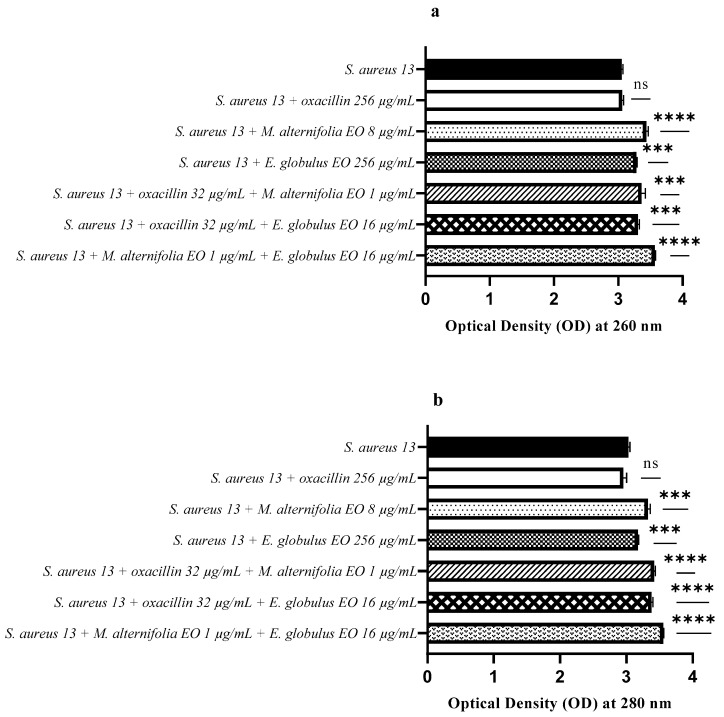
Assessment of nucleic acids and protein release from *Staphylococcus aureus* (MRSA) 13 after treatment with single EO, antibiotic–EO, and EO–E. (**a**) Optical density (OD) at 260 nm demonstrating nucleic acid release, and (**b**) optical density (OD) at 280 nm demonstrating protein release. Each experiment was replicated three times. *p*-values of <0.05 (*), *p* < 0.01 (**), *p* < 0.001 (***) and *p* < 0.0001 (****) were considered significant according to a *t*-test and an ANOVA. ns stands for not statistically significant.

**Figure 11 antibiotics-12-00846-f011:**
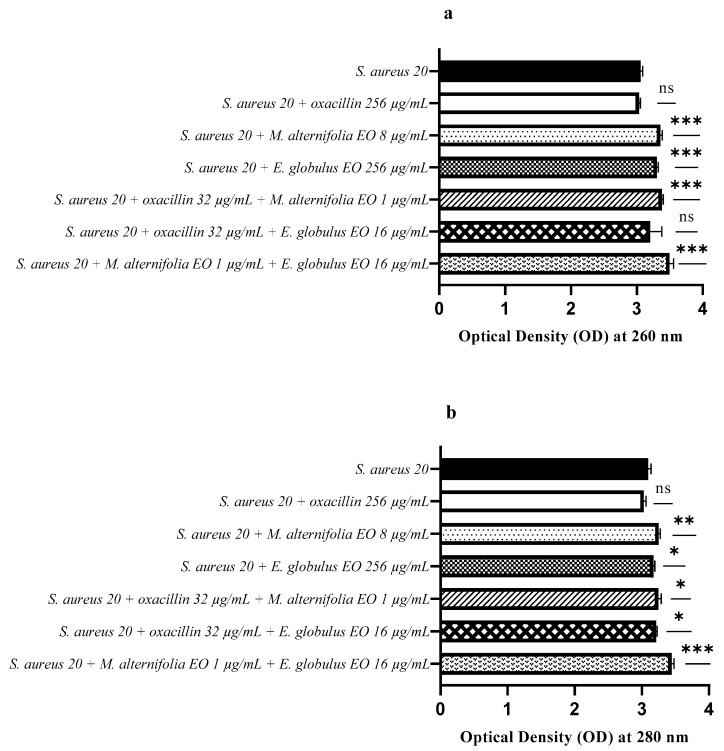
Assessment of nucleic acids and protein release from *Staphylococcus aureus* (MRSA) 20 after treatment with single EO, antibiotic–EO, and EO–E. (**a**) Optical density (OD) at 260 nm demonstrating nucleic acid release, and (**b**) optical density (OD) at 280 nm demonstrating proteins release. Each experiment was replicated three times. *p*-values of <0.05 (*), *p* < 0.01 (**), *p* < 0.001 (***) and *p* < 0.0001 (****) were considered significant according to a *t*-test and an ANOVA. ns stands for not statistically significant.

**Figure 12 antibiotics-12-00846-f012:**
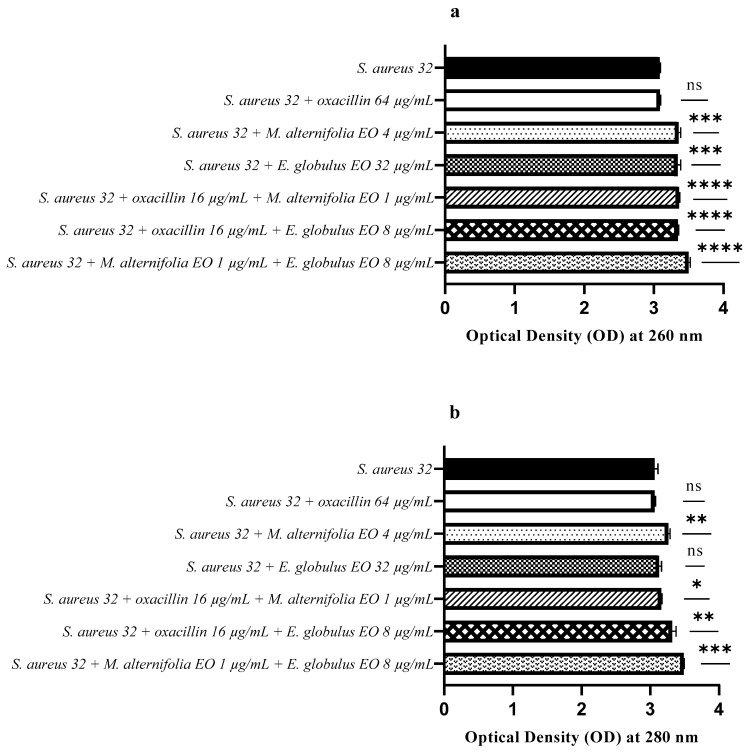
Assessment of nucleic acids and protein release from *Staphylococcus aureus* (MRSA) 32 after treatment with single EO, antibiotic–EO, and EO–E. (**a**) Optical density (OD) at 260 nm demonstrating nucleic acid release, and (**b**) optical density (OD) at 280 nm demonstrating proteins release. Each experiment was replicated three times. *p*-values of <0.05 (*), *p* < 0.01 (**), *p* < 0.001 (***) and *p* < 0.0001 (****) were considered significant according to a *t*-test and an ANOVA. ns stands for not statistically significant.

**Figure 13 antibiotics-12-00846-f013:**
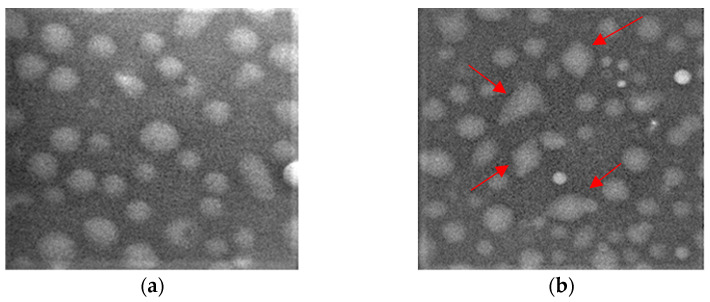
Example of SEM images of a *S. aureus* strain (**a**) untreated and (**b**) treated with the *M. alternifolia* EO–*E. globulus* EO combination at the FIC index concentration. Arrows indicate morphological changes in the *S. aureus* strain caused by EO activity.

**Table 1 antibiotics-12-00846-t001:** Minimum inhibitory concentration (MIC) of oxacillin (µg/mL), *Malaleuca alternifolia*, and *Eucalyptus globulus* EOs (µg/mL) against methicillin-resistant *Staphylococcus aureus* (MRSA) strains.

Strains	Oxacillin (µg/mL)	*M. alternifolia* EO (µg/mL)	*E. globulus* EO (µg/mL)
*S. aureus* 13	256	8	256
*S. aureus* 20	256	8	256
*S. aureus* 32	64	4	32

**Table 2 antibiotics-12-00846-t002:** Synergistic activity of associated EO–EO and EO–antibiotic combinations against methicillin-resistant *Staphylococcus aureus* (MRSA) strains, by fractional inhibitory (FIC) index calculation.

Strains	Oxacillin– *M. alternifolia* EO	Oxacillin– *E. globulus* EO	*M. alternifolia* EO– *E. globulus* EO
*S. aureus* 13	0.25	0.19	0.19
*S. aureus* 20	0.25	0.19	0.19
*S. aureus* 32	0.5	0.5	0.5

**Table 3 antibiotics-12-00846-t003:** Polymerase chain reaction primer used for the amplification of the *mecA* gene detected in this study.

Primer	Nucleotide Sequence (5′ to 3′)	Product Size (bp)
*mecA*	5′-CCTAGTAAAGCTCCGGAA-3′ 5′-CTAGTCCATTCGGTCCA-3′	314

## Data Availability

Not applicable.

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
