# Peer review of "Synergistic Inhibition of Methicillin-Resistant Staphylococcus aureus (MRSA) by Melaleuca alternifolia Chell (Tea Tree) and Eucalyptus globulus Labill. Essential Oils in Association with Oxacillin"

_antibiotics, 2023, doi:10.3390/antibiotics12050846_

Round 1

Reviewer 1 Report

The manuscript 'Synergistic inhibition of methicillin-resistant Staphylococcus aureus (MRSA) by Melaleuca alternifolia Chell (Tea Tree) and Eucalyptus globulus Labill. Essential Oils in association with Oxacillin' is devoted to essential oils (EO) as adjuvants for antimicrobial therapy. The article is concerned with extremely important problem of antibiotic resistance, nonetheless there are some questions about the design of the research.

Major issues:

1. The article demonstrates the point that essential oils damage the membrane. Nonetheless, this fact was known previously, and the corresponding citations are provided in the introduction section (e.g. 0.1007/s00284-018-1594-x). If the aim of the study is verification of this fact (as stated in Line 70), novelty of the study is questionable.

2. The design of the study needs to be clarified. Three clinical isolates were used for testing, how these strains were selected?

3. There is no clear reasoning for choosing oxacillin as an antibiotic for synergy testing. If there is an idea that EO are capable to overcome resistance, it has to be clarified. Moreover, the resistance mechanism in the selected strains has to be discussed.

4. Sections 2.3, 2.4 and 2.5 are devoted to various methods for illustration of the same phenomenon - membrane permeabilization. The raw data on the methods make the results hard to comprehend, so it has to be moved to supplementary materials. Instead of three different sections, the data has to be analyzed and provided in comparative diagrams. Moreover, the detailed description and comparison of the utilized methods is needed.

Minor issues:

1. The authors misleadingly use both fractional inhibitory concentrations (FIC) and fractional inhibitory concentration index (FIC Index). For example, in Table 2 caption mentions FIC Index, but in the table there are FIC values. Individual MICs in this Table makes the whole picture incomprehensible. Moreover, formula for FIC and FIC Index should be provided.

2. The authors suggest combination of EO with antibiotics as a valid trarapeutic alternative (Line 307), but how such combination might be applied for systemic administration?

3. What does the term "Outer membrane" mean if gram-positive MRSA was studied?

The text of the article contains many spelling and grammatical errors, often making it very difficult to understand.

Reviewer 2 Report

The manuscript investigated the antimicrobial activity of Melaleuca alternifolia and Eucalyptus globulus essential oils (EOs) against three strains of methicillin-resistant Staphylococcus aureus (MRSA). The authors demonstrated that the treatment with EO-oxacillin combinations and EO-EO association resulted in a synergistic effect in most of the tests performed. Besides, EO/EO association showed a high activity in the alteration of the membrane, increasing the permeability of about 80% in all the MRSA strains treated. The authors are recommended to elaborate on the differences between this study and another article (10.3390/molecules28041671). The authors are referred to for a more explicit discussion of the importance of oxacillin/EO synergistic antimicrobials. The authors are advised to cite the following articles (10.1016/j.cej.2021.131919; 10.1186/s42825-022-00106-2; 10.1016/j.actbio.2022.04.043; 10.1002/smll.202206220) on the use of non-antibiotic therapies to overcome bacterial resistance. 

No modifications suggested.

Reviewer 3 Report

In the introduction the authors should address in more detail the literature of the plant species of the study.

In the introduction, the authors should address bacterial resistance and mention the resistance mechanisms of antibiotics, for example the mechanism of resistance to oxacillin.

Please add resistance profile of used clinical isolates.

The chemical composition of essential oils must be mentioned (GC-MS Analysis). This, in turn, is one of the conditions for the manuscript to be accepted.

Figures 4, 5, 6 ,10, 11 and 12 should be improved in identifying the type of test presented.

In figure 10,11 add 12 the significance in figure 10 add the significance.

The discussion should focus on the chemical composition to explain the observed antibacterial activity and the observed synergism.

The image in electronic microscopy should be better.   

Reviewer 4 Report

Even your EOs were purchased from a local herbalist’s shop in Modena, Italy. and GC-MS analysis in Molecules. 2023, 28, 1671.

In my opinion, Tea tree planting place, and more discussion about the EOs Chemical configuration with the antibiotic relationship are essential and needed.

Minor editing of English language required

Round 2

Reviewer 1 Report

The revised version of the manuscript contains an improved descriptions of the methods. Novelty of the study is still not evident, as most of the results replicate previous studies on essential oils. Nonetheless, the manuscript might be interesting for the readership of Antibiotics.

Minor points:

1. The title of the section 4.6 still contains the 'outer membrane' term

2. The possible applicability of essential oil in the treatment of wounds (the answer to the minor point 2 in the previous review) should be discussed in the manuscript

Reviewer 3 Report

Dear editor, all corrections were made and the manuscript was accepted.
